# Microsatellite-Based Genetic Diversity Analysis and Population Structure of Proso Millet (*Panicum miliaceum* L.) in Kazakhstan

Meisam Zargar [1,*], Elmira Dyussibayeva [2], Aidyn Orazov [3], Aiym Zeinullina [2], Irina Zhirnova [2], Gulzat Yessenbekova [2] and Aiman Rysbekova [2,*]

1   Department of Agrobiotechnology, Institute of Agriculture, RUDN University, 117198 Moscow, Russia
2   Agronomy Faculty, S. Seifullin Kazakh AgroTechnical Research University, Astana 010011, Kazakhstan; elmira_dyusibaeva@mail.ru (E.D.); aiym._92@mail.ru (A.Z.); ira777.89@mail.ru (I.Z.); gulzat_es@mail.ru (G.Y.)
3   Higher School of Natural Sciences, Astana International University, Astana 010000, Kazakhstan; orazov_aidyn@mail.ru
*   Correspondence: zargar_m@pfur.ru (M.Z.); aiman_rb@mail.ru (A.R.)

**Abstract:** Proso millet is an important allotetraploid cereal crop; however, it is the least studied species of the *Poaceae* family, as it is an under-utilized crop. Genomic resources for proso millet are very limited compared to major crops. An understanding of the genetic relationships among germplasm resources is important for future breeding endeavors. In the present study, simplesequence repeat (SSR) markers were employed to assess the polymorphism and genetic diversity of 100 millet accessions from different countries, which were tested in the dry steppe zone conditions of the Akmola region from 2020 to 2022. The use of 20 SSR markers detected a total of 47 alleles, with an average allele number of 2.35 per locus among these proso accessions. Nine of them were polymorphic among the genotypes, which suggests that these SSR markers can be used for genetic studies. The results showed a moderate level of polymorphism information content (PIC) that averaged at 0.424, ranging from 0.125 to 0.795. The markers SSR-67, SSR-82, SSR-85 and SSR-109 showed high PIC values of 0.536, 0.756, 0.795 and 0.758, respectively. Markers SSR 85 and SSR 86 significantly correlated to agronomic traits, such as productive tillering (PT) and grain yield (GY). The genetic structure, UPGMA cluster and PCoA assay indicated that the accessions that originated from Central Asia had higher genetic diversity. Based on structure (K = 3), all the accessions were divided into three groups, where the gene pool that originated from Central Asia wasdetected in all three clusters. Based on a principal component analysis (PCA), the accessions of Central Asian origin were genetically closer to the North Asian group.

**Keywords:** proso millet collection; genetic diversity; SSR markers; agronomic traits



## 1. Introduction

Proso millet (*Panicum miliaceum* L.), also known as common millet (USA) or broomcorn millet (China), belongs to the genus *Panicum*, the tribe *Paniceae*, the family *Poaceae* and the order *Poales* [1]. Millet is a small-grained self-pollinated allotetraploid cereal crop and has a short growing season. The plant was domesticated about 10,000 years ago in Central and Eastern Asia [2,3]. Proso millet is (2n = 4× = 36) an annual herbaceous plant and currently cultivated in Asia, Australia, North America, Europe and, more rarely Africa [4–6]. Asian countries use millet as a food crop, and the United States actively grows the plant to produce valuable fodder for birds and livestock [7].

Although the grain yield of proso millet is lower than that ofmajor crops, it is highly adapted to a semi-arid climate [8]. Today, the global production of millet is at a fairly high level and occupies a significant area. According to the FAO, millet ranks 6th among cerealsworldwide in terms of its cultivated area (34.7 million hectares) and gross grain harvest (31.6 million tons), following wheat, rice, barley, corn and sorghum. Proso millet is cultivated in 30 countries [9,10], on about 0.82 million ha of land in Russia, 0.32 million ha

in China [11], 0.20 million ha in the USA [12], 0.03 million ha in India [13] and 0.002 million ha in Korea [14].

Inancient times, millet was worshipped by the people of Central Asia. So, in the diet of the Kazakhs, there have always been such foods as tary and talcane, which are suitable for long-term storage and providing valuable nutritional elements in a nomadic way of life. Now, in the Republic of Kazakhstan, proso millet is one of the main cereal crops, owing to its high drought resistance, salt resistance and lesser dependence on the sowing dates of the drysteppe zone. Due to the development of the virgin lands of the country, the sown area of millet reaches 1.7 million hectares. The largest areas that are designated for the cultivation of this crop are in the agricultural regions: Pavlodar, Akmola, Aktobe, West Kazakhstan (Ural) and Kostanay. In Western Kazakhstan, millet is the main cereal crop [15]. Millet grain contains all the essential amino acids, 10 to 15% protein, 55–65% starch, more than 5% fat, 0.3–0.9 mg% carotenoids and a relatively small amount of fiber. Millet contains various vitamins such as PP, $B_1$ and $B_2$, as well as large amounts of potassium, magnesium, phosphorus, molybdenum, magnesium, iodine, zinc, sodium and bromine [16–18].

Assessments of genetic diversity inGenBankcollections has significance for plant breeding programs and genetic resource conservation [19–23]. The proso millet genome is relatively small and sized at 923 megabases [24,25]. Population structure studies of agricultural cropgene pools based on genotyping is important for the genetic improvement of economically valuable traits [26].

Currently, the range of markers used for molecular genetic analysis is very extensive, but studies on genetic diversity based on the molecular markers of worldwide collections of proso millet are limited [27]. Molecular studies of the DNA polymorphism of the proso millet germplasm are mainly based on the following markers: RAPD [28,29], ISSR [30,31], AFLP [32] and SSR [6,33]. Among all PCR-based molecular markers, simple sequence repeat (SSR) markers have been proven to be the most widely used in germplasm collection characterization. They are highly reproducible, co-dominant, multi-allelic, numerous, have a relatively low cost, are evenly distributed in the plant genome and present a high degree of polymorphism and information value [34,35]. Most of the SSR markers in proso millet were developed using the genomic resources of related grass species. Hu et al. identified 46 SSR markers from rice, wheat, oats and barley to analyze the genetic diversity of millet [36]. Cho et al. (2010) developed 25 SSR markers from a proso millet BAC library [37], but they have not yet been widely utilized with *P. miliaceum*. [38]. Currently, about 100 SSR markers are available for genetic and breeding studies of millet. A population-level analysis was conducted by Rajput and Santra; 100 SSR markers were used on 90 samples of millet, and the results discovered a relationship between the genetic clustering and geographic origin [6]. The first case of breeding of waxy proso millet in Kazakhstan was presented by Zhirnova et al. [39]; the dCAPS (9bF/15delRB) molecular marker was applied. There is limited information on SSR marker-based genetic diversity analyses of the Kazakhstan GenBank, which is important for its genetic improvement [40]. The objectives of this study were to evaluate the genetic diversity in a proso millet collection consisting of 100 accessions from different ecological and geographical zones using SSR markers, and to identify SSR markers associated with the main agronomic traits to improve millet breeding in the future.

## 2. Materials and Methods

### 2.1. Plant Material

A total of 100 proso millet (*Panicum miliaceum* L.) varieties and accessions of various ecological and geographical origins served as the experimental material. Approximately 50 accessions were sourced from the germplasm collection of the Peginal Plant Introduction Station (the world collection of the USDA), Iowa State University (USA). The germplasm collection was divided into 6 groups based on origin: American, European, East Asian, Southwest Asian, Central Asian and North Asian (Figure 1).

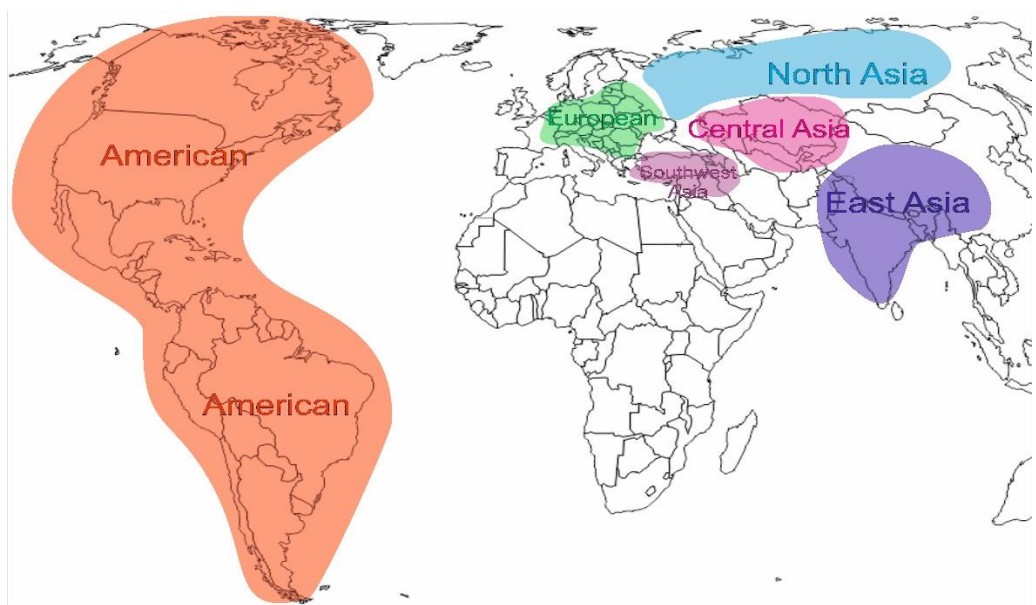

**Figure 1.** Geographical distribution of accessions divided into 6 groups based onorigin. American—4; European—8; East Asian—10; Southwest Asian—11; Central Asian—26; North Asian—41.

The experiments were carried out in a laboratory as well as in the field. The laboratory studies were performed at the Scientific Research Platform of Agricultural Biotechnology (RPAB) at the Saken Seifullin Kazakh Agrotechnical Research University, Astana, Republic of Kazakhstan.

For the identification of molecular markers that are associated with the main agronomic traits, field experiments were conducted in the plant nursery of the A.I. Baraev Scientific Production Center of Grain Farming (Shortandyvillage-1, Shortandy district, Akmola region, Kazakhstan) in the dry steppe zone of the Akmola region (51°41′58″ N; 70°59′41″ E; DD: 51.6995, 70.9946; height above mean sea level in meters: 375). The field observations and the collection characterization were conducted from May to September for three consecutive years, i.e.,the 2020, 2021 and 2022 growing seasons (Table 1).

**Table 1.** Meteorological conditions and characteristics for the experimental site during the agronomic seasons of 2020, 2021 and 2022.

| Months | Mean Temperature T, °C | | | Mean Precipitation, mm | | |
|---|---|---|---|---|---|---|
| | Per Month, 2020 | Per Month, 2021 | Per Month, 2022 | Per Month, 2020 | Per Month, 2021 | Per Month, 2022 |
| May | 17.8 | 17.2 | 15.7 | 1.0 | 12.1 | 16.9 |
| June | 15.8 | 18.4 | 20.2 | 50.1 | 18.3 | 22.2 |
| July | 17.7 | 20.4 | 21.1 | 46.6 | 31.9 | 52.9 |
| August | 19.6 | 19.6 | 17.2 | 27.3 | 37.8 | 25.2 |
| September | 10.9 | 10.2 | 13.2 | 32.2 | 40.5 | 8.0 |

The experiment was performed according to the All-Russian Institute of Plant Growing guidelines and the field experiment methodology [41]. The nursery collection was sown in a crop rotation selectionfollowing the predecessor of spring wheat. The sowing of the nursery collection was carried out with a selection seeder duringthe lastten days of May; the plot area was 1 m², and the location of the plots was systematic, with every tenth number being of the standard variety Saratovskoe 6. The row spacing was 20 cm, the distance between the plants was5 cm and the seeding depth was 5 cm.

Observations of the yield and yield contributing traits were recorded for each accession and for every replication: seed weight per panicle (g), thousand seed weight (g), productive

tillering (pcs) and grain yield (g/m$^2$). The number of seeds was counted on the "DATA Count S-25".The thousand seed weight was calculated as the average of 10 replicates containing 100 seeds, multiplied by 10. The difference between the 1000 seed weights was obtained by a counter, upto themaximum level [42].

### 2.2. DNA Extraction

The leaves of the proso millet germplasm seedlings were collected and used for the isolation of DNA for SSR marker genotyping. The genomic DNA from each genotype was isolated from 7-day-old chlorophyll-free seedlings following a modified cetyl trimethylammonium bromide (CTAB) method [43]. The concentration of the genomic DNA samples was determined with a UV-Vis spectrophotometer (NanoDrop 2000, Thermo Fisher Scientific, Waltham, MA, USA). The integrity of the DNA was evaluated viaelectrophoresis on 1% agarose gel.

### 2.3. SSR Marker Analysis

The genotyping of the proso millet germplasm was performed using a set of twenty (20) SSR markers (Table 2).

**Table 2.** The list of SSR primers used for screening the proso millet collection [44].

| Marker | Forward 5′-3′ | Reverse 5′-3′ | Annealing Temperature, ($^\circ$C) |
|---|---|---|---|
| SSR 67 | ACTAGGTAATTACAGGGGAG | GGCATGTGGAGTAGTAGTAT | 46 |
| SSR 70 | ACTCATCTGACAAACTATGG | ATAGAACTGTGTGTTGGTGT | 45 |
| SSR 71 | ACTCATGATTAAAGGGTGAT | TGTGACAACATTGTGAATAG | 46 |
| SSR 82 | ACCAGCCCCAACTAC | ATTGTTTATGTGATCTCAGG | 45 |
| SSR 85 | ACCAGTACGGCAACC | ATTTCTCTTTGATCTTCTCC | 45 |
| SSR 86 | ACCAGTACGGCAACC | TTGATCTTCTCCTTAATGC | 45 |
| SSR 92 | ACCCACCCAACCAGT | TACTTTGTCCTTTTCCAGTA | 46 |
| SSR 100 | ACCTAGACAAATGCGTACT | CAAAACCAAACCCTCTC | 45 |
| SSR 109 | ACCTTAAGGATTGGAATATC | GTTGAGTAAGTTTCTCCTCA | 46 |
| SSR 120 | ACGACCATGATCTCATAAC | GAGGATGATGAGTAGGAAGT | 45 |
| SSR 121 | ACGACGATGATGATGAC | TCTGGTCAAGTACTCAATTC | 46 |
| SSR 127 | ACGAGGAGATGGATCAG | CTCTCTGTCCGTGGTC | 46 |
| SSR 128 | ACGATGATGAAGAAGCA | GAACTGGCAGAAGCAC | 46 |
| SSR 129 | ACGATGGGGTCTACG | AGCTTAACCCTGAACTTCT | 45 |
| SSR 131 | ACGCAGCCTCATCAT | TAAGAAGCTGAGATTTGGT | 45 |
| SSR 142 | ACTAAGAGGAAGCCTATGTT | AACTGCAGCTACATTGTATT | 45 |
| SSR 143 | ACTAAGAGGAAGCCTATGTT | TACAGCAGTGCAGATATTTA | 45 |
| SSR 144 | ACTAAGAGGAAGCCTATGTT | TTAAGCTGGAAAGTAATCAG | 45 |
| SSR 146 | ACTACAAGAGCAAGTCCAC | AAATACAACATTGCAAGACT | 45 |
| SSR 182 | ACAACAGATTTCTAAACCAA | TCTCGGAGAACATCAAG | 45 |

The PCR amplifications were performed in a total volume of 15 μL, containing 8 μL of 2×Master Mix for PCR (BioRad, Hercules, CA, USA), 5.2 μLof ddH$_2$O, 10 μM and 1 μL of each primer (F, R) (*Lumiprobe* Corporation (USA)) and 100–150 ng of the DNA template. The PCR amplification was run using a VeritiPro™ Thermal Cycler (Applied Biosystems, Singapore) with the following program: denaturation at 95°C for 5 min, followed by 35 cycles of 40 sec at 95°C, 40 s at the annealing temperature, 35 sec at 72°C, and finally anextension at 72°C for 10 min. The PCR products were separated on a 6% polyacrylamide gel followed by ethidium bromide, and visualized using a gel documenting system (DocPrint CX3, Vilber, Paris, France, 2020). The marker alleles for all SSRs were recorded manually using the PAAG gels. The SSR fragments were assessed for the presence of bands in the gel profile. When two DNA bands of different sizes were observed at least in two genotypes, the marker was considered polymorphic. The DNA fragments of different sizes were considered different alleles, while DNA fragments of same size were considered as the same allele.

### 2.4. Data Analysis

The obtained field data wereanalyzed usingMicrosoft Excel software2016version 16.0.4266.1001. Based on the SSR scoring data, calculations for the number of different alleles (Na), the number of effective alleles (Ne), Shannon's information index (I), the expected heterozygosity (He), the unbiased expected heterozygosity (uHe) and the polymorphic information content (PIC value) were performed using the GenAlEx 6.5 application for MS-Excel [45,46]. The population structure analysis was carried out using STRUCTURE 2.2 software [47]. The number of subgroups (K) in the population was determined by running the software with 10 independent replicate runs per K value (number of clusters), ranging from 1 to 10. Each run involved a burning period of 100,000 iterations, and a postburning simulation length of 1,000,000. The principal coordinate analysis (PCoA) was performed using GenAlEx 6.5. A neighbor-joining phylogenetic tree was constructed usingthe unweighted pair group method with arithmetic average (UPGMA) using PAST v.3.25 software [48].

## 3. Results

### 3.1. Agronomic Traits in the Studied Groups of Proso Millet by Origin

Under the climate conditions of North Kazakhstan, proso millet is typically planted in late May or early June and harvested in early September. In general, four agronomic traits—the seed weight per panicle (SWP, g), the thousand seed weight (TSW, g), the productive tillering (PT, pcs) and the grain yield (GY, g/m$^2$)—were studied. The values of the minimum/maximum (max/min), range, mean, coefficient of variation (CoV) and standard deviation (SD) for the four traits that were under study for each origin group arepresented in Table 3.

**Table 3.** Average agronomic traits in the proso germplasm according to the three-year field trial data.

| Origin Group | Descriptive Statistics | SWP, g | TSW, g | PT, pcs | GY, g/m$^2$ |
|---|---|---|---|---|---|
| American | Min/max | 2.1/2.8 | 5.8/7.4 | 1.1/1.3 | 342.0/708.0 |
| | Range | 0.7 | 1.6 | 0.2 | 366.0 |
| | Mean | 2.6 | 6.6 | 1.2 | 474.2 |
| | CoV | 26.9 | 24.2 | 16.6 | 77.1 |
| | SD | 0.4 | 0.6 | 0.05 | 150.7 |
| European | Min/max | 1.5/3.3 | 4.5/7.1 | 1.1/1.2 | 334.0/678.0 |
| | Range | 1.8 | 2.6 | 0.1 | 344.0 |
| | Mean | 2.8 | 5.4 | 1.1 | 438.0 |
| | CoV | 64.2 | 48.1 | 9.0 | 78.5 |
| | SD | 0.5 | 0.7 | 0.04 | 31.0 |
| East Asia | Min/max | 1.7/3.1 | 4.4/7.2 | 1.1/1.3 | 244.0/471.0 |
| | Range | 1.4 | 2.8 | 0.2 | 227.0 |
| | Mean | 2.7 | 5.3 | 1.2 | 320.5 |
| | CoV | 51.8 | 52.8 | 16.6 | 70.9 |
| | SD | 0.6 | 1.2 | 0.03 | 48.7 |
| Southwest Asia | Min/max | 1.7/3.8 | 4.9/6.4 | 1.1/1.5 | 275.0/706.0 |
| | Range | 2.1 | 1.5 | 0.4 | 431.0 |
| | Mean | 2.6 | 5.9 | 1.2 | 439.3 |
| | CoV | 80.7 | 25.4 | 33.3 | 98.1 |
| | SD | 0.7 | 0.4 | 0.04 | 160.0 |
| Central Asia | Min/max | 1.1/7.5 | 3.7/7.8 | 1.1/1.6 | 179.0/1248.0 |
| | Range | 6.4 | 4.1 | 0.5 | 1069.0 |
| | Mean | 2.6 | 6.5 | 1.2 | 440.7 |
| | CoV | 246.1 | 63.0 | 41.6 | 242.5 |
| | SD | 0.6 | 0.5 | 0.07 | 127.0 |

**Table 3.** *Cont.*

| Origin Group | Descriptive Statistics | SWP, g | TSW, g | PT, pcs | GY, g/m$^2$ |
|---|---|---|---|---|---|
| | Min/max | 1.4/3.9 | 3.7/7.5 | 1.0/1.5 | 225/697 |
| | Range | 2.5 | 3.8 | 0.5 | 472 |
| North Asia | Mean | 2.8 | 6.1 | 1.3 | 535.2 |
| | CoV | 89.2 | 62.2 | 38.4 | 88.1 |
| | SD | 0.7 | 0.6 | 0.1 | 105.9 |

Note: SWP—seed weight per panicle, TSW—thousand seed weight, PT—productive tillering, GY—grain yield.

A high yield and its related components are important targeted traits forproso millet. Field trials for three years revealed a marked difference in the GY between the studied groups based on their origin. The highest average GY values were revealed for the North Asian and American groups; the average yield accounted for 535.2 and 474.2 g/m$^2$, respectively. The SWP and PT traits did not show significant results based on the groups; the SWP ranged from 2.6 to 2.8 g, and the PT ranged from 1.1 to 1.3. As for the TSW, the highest average values were for the American and Central Asia groups (6.6 and 6.5 g, respectively) and the lowest ones were for the European and East Asian groups (5.4 and 5.3 g, respectively). The coefficient of variation was the highest for the SWP and GY in the Central Asian group: 246.1 and 242.5, respectively.

### 3.2. Genotyping usingSSR Markers of Proso Millet Collection

A set of 20 SSR markers was used to analyze the genetic diversity of 100 millet accessions collected from 16 various ecological and geographical areas. Nine SSR markers out of the 20 SSR markers were polymorphic: SSR 67, SSR 82, SSR 85, SSR 86, SSR 92, SSR 100, SSR 109, SSR 142 and SSR 146; the other 11 were monomorphic. These 20 primers amplified a total of 47 alleles; among them, 31 were polymorphic. The sizes of the observed fragments ranged from 132 to 580 bp (Table 4).

**Table 4.** Details of SSR markers used during the present study.

| Locus | Observed Allele Size in Proso Millet (bp) | Number of Alleles | Number of Polymorphic Bands | Polymorphism, % |
|---|---|---|---|---|
| SSR 67 | 200, 225, 250, 275 | 4 | 4 | 100 |
| SSR 70 | 132 | 1 | 0 | 0 |
| SSR 71 | 191 | 1 | 0 | 0 |
| SSR 82 | 230, 250, 260, 290, 310, 340, 370, 490 | 8 | 8 | 100 |
| SSR 85 | 340, 360, 400, 450, 500, 580 | 6 | 6 | 100 |
| SSR 86 | 300, 360, 430 | 3 | 2 | 67 |
| SSR 92 | 280, 300 | 2 | 1 | 50 |
| SSR 100 | 270, 300 | 2 | 1 | 50 |
| SSR 109 | 180, 200, 220, 350, 400, 580 | 6 | 6 | 100 |
| SSR 120 | 224 | 1 | 0 | 0 |
| SSR 121 | 183 | 1 | 0 | 0 |
| SSR 127 | 266 | 1 | 0 | 0 |
| SSR 128 | 263 | 1 | 0 | 0 |
| SSR 129 | 239 | 1 | 0 | 0 |
| SSR 131 | 349 | 1 | 0 | 0 |
| SSR 142 | 140, 400, 500 | 3 | 2 | 67 |
| SSR 143 | 144 | 1 | 0 | 0 |
| SSR 144 | 450 | 1 | 0 | 0 |
| SSR 146 | 182, 200 | 2 | 1 | 50 |
| SSR 182 | 200 | 1 | 0 | 0 |

A maximum of eight alleles was observed for SSR 82, six alleles for SSR 85 and SSR 109, four alleles for SSR 67, three alleles for SSR 86 and SSR 142, andtwo alleles for SSR 92,

SSR 100 and SSR 146. The other 11 SSRs had one allele (SSR 70, SSR 71, SSR 120, SSR 121, SSR 127, SSR 128, SSR 129, SSR 131, SSR 143, SSR 144 and SSR 182). The highest percentage of polymorphism (100%) was observed inSSR 67, SSR 82, SSR 85 and SSR 109, while SSR 86, SSR 92, SSR 100, SSR 142 and SSR 146 showed a medium level (50–67%) of polymorphism.

Among the 100 proso millet accessions, a total of 47 alleles wereidentified, with an average of 2.35 alleles per marker. The genetic diversity analysis showed that the mean number of different alleles (Na) per SSR locus ranged from 1.77 to 3.66, for which no unique alleles were obtained (Table 5).

**Table 5.** Genetic diversity information of each group, provided by SSR markers.

| Origin | Mean/SE | Na [a] | Ne [b] | I [c] | He [d] | uHe [e] |
|--------|---------|--------|--------|-------|--------|---------|
| American | Mean | 2.111 | 1.904 | 0.573 | 0.347 | 0.397 |
| | SE | 0.351 | 0.339 | 0.170 | 0.097 | 0.111 |
| European | Mean | 2.556 | 2.209 | 0.746 | 0.441 | 0.471 |
| | SE | 0.377 | 0.331 | 0.169 | 0.095 | 0.101 |
| East Asia | Mean | 1.778 | 1.584 | 0.370 | 0.222 | 0.234 |
| | SE | 0.364 | 0.299 | 0.170 | 0.100 | 0.105 |
| Southwest Asia | Mean | 2.000 | 1.636 | 0.461 | 0.282 | 0.296 |
| | SE | 0.333 | 0.281 | 0.147 | 0.086 | 0.090 |
| Central Asia | Mean | 3.667 | 2.153 | 0.827 | 0.450 | 0.456 |
| | SE | 0.624 | 0.334 | 0.155 | 0.074 | 0.075 |
| North Asia | Mean | 3.556 | 2.093 | 0.795 | 0.421 | 0.430 |
| | SE | 0.689 | 0.348 | 0.172 | 0.079 | 0.081 |

Note: [a] number of different alleles; [b] number of effective alleles; [c] Shannon's information index; [d] expected heterozygosity; [e] unbiased expected heterozygosity.

The mean number of effective alleles was higher in the European, Central Asian and North Asian groups (2.209, 2.153 and 2.093, respectively), while it was lower in the American, Southwest Asian and East Asian groups (1.904, 1.636 and 1.584, respectively). The number of effective alleles (Ne) ranged from 1.584 to 2.209, with an average of 1.929. The Shannon's information index (I) ranged from 0.370 to 0.827, with an average of 0.628. The expected heterozygosity (He) ranged from 0.222 to 0.441, with a mean value of 0.360. The average value of the unbiased expected heterozygosity (uHe) was 0.380, with values of 0.234 for the East Asian group, 0.296 for Southwest Asian, 0.397 for American, 0.430 for North Asian, 0.456 for Central Asia and 0.471 for the European group, respectively.

The average of the polymorphic information content (PIC) value generated by the SSR primers was moderate at 0.424, ranging from 0.125 for SSR-67 to 0.795 for SSR-86 (Table 6).

**Table 6.** Polymorphic information content (PIC value) of each SSR markers.

| Markers | PIC |
|---------|-----|
| SSR-67 | 0.536 |
| SSR-82 | 0.756 |
| SSR-85 | 0.795 |
| SSR-86 | 0.125 |
| SSR-92 | 0.158 |
| SSR-100 | 0.158 |
| SSR-109 | 0.758 |
| SSR-142 | 0.278 |
| SSR-146 | 0.256 |
| Mean | 0.424 |
| SE | 0.277 |

Out of nine SSR markers, four showed high PIC values: SSR 67 (0.536), SSR-82 (0.756), SSR-85 (0.795) and SSR-109 (0.758); thisindicates that they exceedthe critical value of 0.5.

The evaluation of the amount of genetic variation in the proso millet germplasm is an important task for breeding. A t-test analysis was used to detect the correlations between the SSR markers and the studied agronomic traits (Table 7).

**Table 7.** Correlation analysis of SSR markers associated with agronomic traits.

| Markers | SWP, g | TSW, g | PT, pcs | GY, g/m$^2$ |
|---|---|---|---|---|
| SSR-67 | 0.850 | 0.684 | 0.187 | 0.212 |
| SSR-82 | 0.456 | 0.344 | 0.630 | 0.293 |
| SSR-85 | 0.452 | 0.791 | 0.013 * | 0.029 * |
| SSR-86 | 0.685 | 0.158 | 0.008 * | 0.232 |
| SSR-92 | 0.299 | 0.864 | 0.296 | 0.117 |
| SSR-100 | 0.469 | 0.700 | 0.410 | 0.118 |
| SSR-109 | 0.983 | 0.382 | 0.202 | 0.661 |
| SSR-142 | 0.165 | 0.413 | 0.210 | 0.431 |
| SSR-146 | 0.318 | 0.880 | 0.182 | 0.270 |

*A *p*-value of 0.05 or lower is generally considered statistically significant.

Markers SSR 85 and SSR 86 showed a significant relationship to variance in the PT trait; the *p*-value was 0.013 and 0.008, respectively. Marker SSR-85 wasalso associated with the mean grain yield, at *p*< 0.05.

### 3.3. Population Structure, UPGMA Cluster and PCoA

The relationships among the genotypes from different origin groups based on the SSR genotyping of 100 accessions was determined viagenetic structure analysis, principal coordinates analysis (PCoA) and UPGMA clustering analysis. A structure analysis of the proso millet germplasm was performed using the model-based software program STRUCTURE 2.3.1,and was evaluated using the results from K = 2 to K = 10. Each accession in the core collection was grouped to a specific cluster based on its K value that resulted from the population structure analysis (Figure 2).

According to the K = 2 model, the proso millet germplasm were differentiated into two clusters; the first one consisted mostly of groups that originated from Central Asia, North Asia and Europe. The second cluster, in general, included the East Asian, Southwest Asian and American groups. Some of the accessions of different origins were assigned to both the first and second clusters with different degrees of probability, which indicates their intermediate position. The overall assignments of the genotypes for clusters 1 and 2 were 52% and 48%, respectively. At K = 3, we observed that cluster 1 (red color), cluster 2 (green color) and cluster 3 (blue color) contained 22, 43 and 35 accessions, respectively. The cultivars from the American, European and Southwest Asian groups were mostly assigned to the clusters 1 and 3, while the East Asia group accessions were mostly included in clusters 2 and 3. The accessions from Central Asian and North Asian groups were detected in all three clusters. Similarly, at K = 4, the same phenomenon occurred as at K = 3. At K = 5, most accessions of the East Asian group were detected in cluster 1; the gene pool of the Southwest Asian group in cluster 2 and genotypes of the European group weremainly included in cluster 3. Genotypes from the Central Asian and North Asian groups dominated cluster 4; these groups were also distributed all over the cluster. Cluster 1 had 14 genotypes, cluster 2 had 22 genotypes, cluster 3 had 31 genotypes and cluster 4 had 23 genotypes. Only 10 genotypes from the European, Southwest Asian, Central Asian and North Asian groups were included in cluster 5. The genotypes of American origin were included in clusters2 and 4.

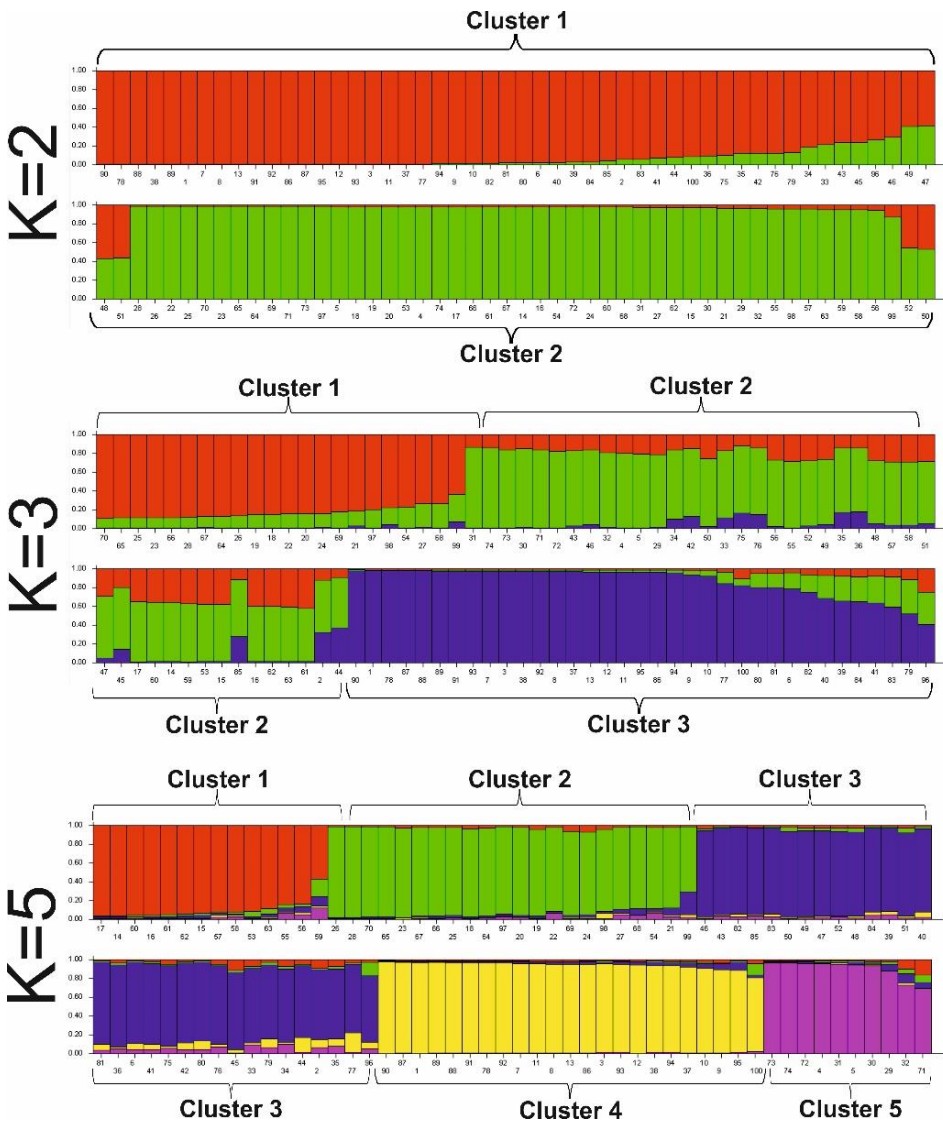

**Figure 2.** Population structure analysis results of 100 proso millet accessions: clustering diagram at K = 2, at K = 3 and at K = 5.

All 100 proso millet accessions were discriminated successfully using theirSSR markers based on the UPGMA (Figure 3).

The results of the UPGMA clustering based on the similarity index found that the 100 proso millet accessions were divided into three groups (Groups 1, 2 and 3). Group 1 (blue color) included 53 accessions, while Group 2 (red color) consisted of 32 accessions and Group 3 (green color) of 15 accessions. The first group mostly included the genotypes from Central Asia (24), North Asia (21), Europe (6), Southwest Asia (1) and America (1). The second cluster contained accessions from Southwest Asia (10), Central Asia (7), East Asia (6), North Asia (4), America (3) and Europe (2), while the third group mainly included germplasm from Central Asia (10), East Asia (4) and North Asia (1). Group 1 was further subdivided into three subgroups. Subgroup I consisted of 22 genotypes, which belonged to Central Asia, North Asia and America. Subgroup II comprised 19 genotypes; most of them belonged to Central Asia, North Asia and Europe. Twelve cultivars were categorized into subgroup III, which constituted Central Asia, North Asia, European and Southwest Asia. Although the accessions from the Central and North Asian groups dominated the first cluster, some varieties of these origins were distributed inall three clusters.

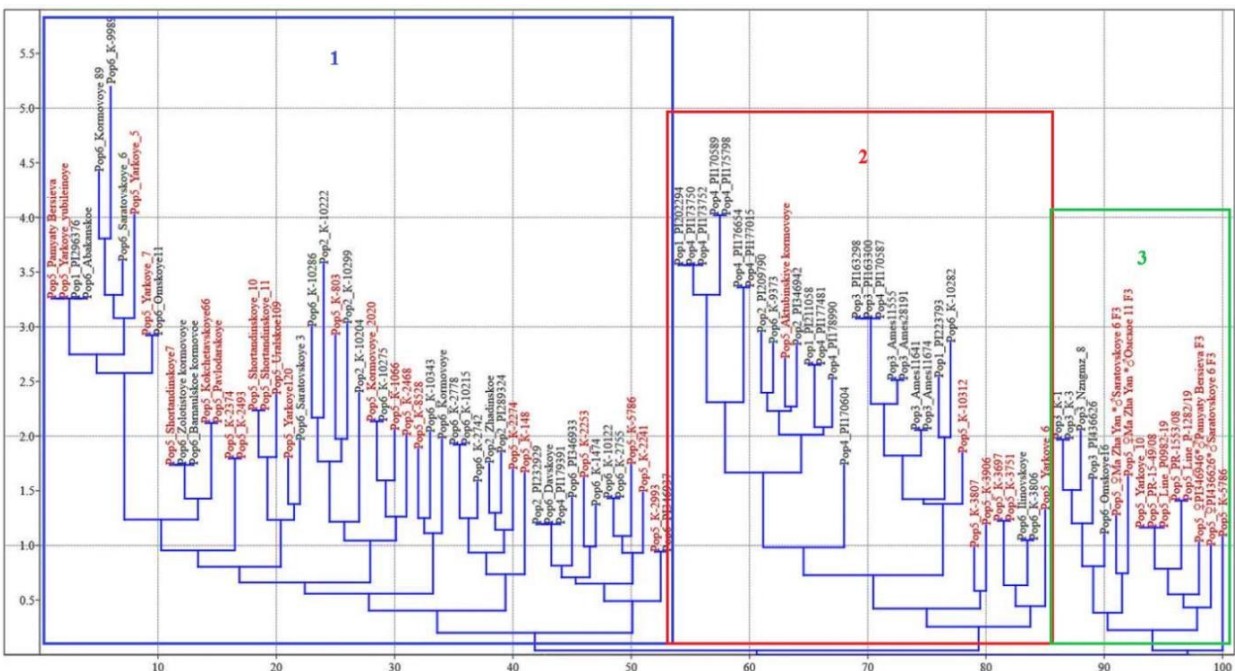

**Figure 3.** UPGMA dendrogram of proso millet germplasm accessions based on 47 alleles generated using 20 SSR markers.

In order to obtain an alternative view of the phylogenetic relationships among the 100 proso millet accessions, a PCoA was performed. The principal coordinates accounted for variation values of 15.77 and 12.20, respectively (Figure 4).

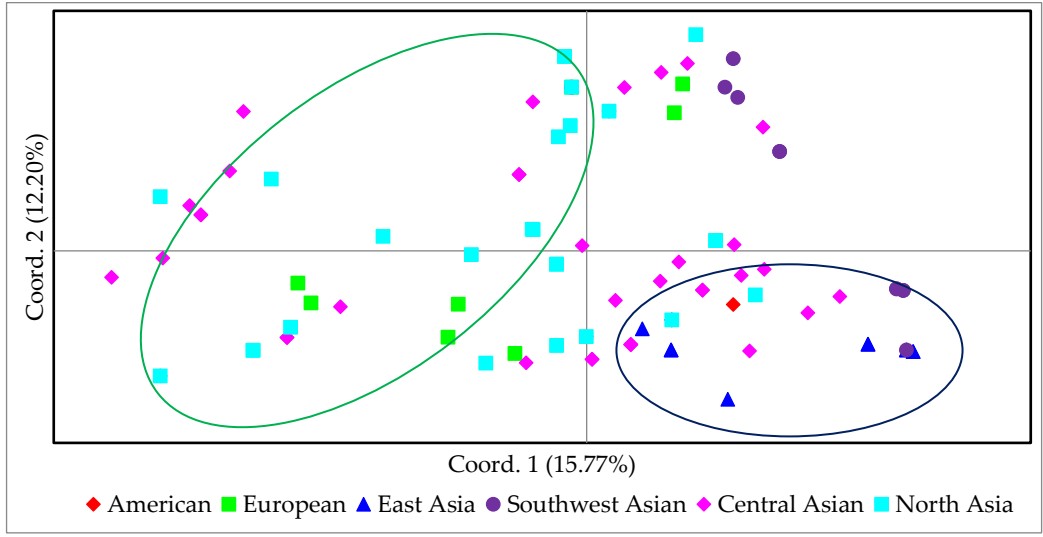

**Figure 4.** Principal component analysis of the proso millet germplasm based on SSR markers.

A score plot of Coord.1 and Coord.2 showed the relationships among the 100 accessions. The PCoA clearly separated the East Asian group from the others. The majority of the East Asia accessions clustered together on the bottom right of the plot. The Southwest Asian genotypes were located alongside the accessions from Central and Eastern Asia (top and bottom right). The European and American groups presented an intermediate position between the Central and North Asian groups. The accessions from Central Asia and Northern Asia were distributed across the whole spectrum, which was similar to the pattern shown in the population structure and UPGMA dendrogram analyses. The PCoA using the

origin data revealed that most of the genotypes of Central Asian origin were genetically closer to the North Asian group.

## 4. Discussion

There is much research on the use of molecular markers in assessing thegenetic diversity of main agricultural crops in Kazakhstan, such as wheat [49], barley [50], rice [51], soybean [52], etc. Despite the importance of proso millet, the available information onits genetic diversity, genetic relationships, phylogenetic relationships, population structure and core collection are still limited. There are many conclusions regarding the genetic diversity of proso millet based on AFLP [32], RAPD [53], intersimple sequence repeat, single nucleotide polymorphism [38] and SSR [37] molecular marker analyses. It was discovered that the proso millet accessions demonstrated high polymorphism levels and grouped together according to their geographical origins based on their RAPD markers [53]. A genetic diversity analysis was performed by Santosh G. et al. using 100 simplesequence repeat (SSR) markers in the United States proso millet genotypes (landraces and cultivars). Highly genetically diverse proso millet hasbeen found in the US germplasm [37]. In the present work, four agronomic traits, such as SWP(g), TSW(g), PT(pcs) and GY(g/m$^2$) were studied. All of the studied accessions of the proso millet collection wereseparated according to their species classification and origin. We analyzed the genetic diversity between different origins of proso millet using SSR markers. Twenty pairs of SSR primers were used for the initial analysis of the proso millet germplasm and nine of them demonstrated polymorphism. The percentage of polymorphic bands across the SSR markers varied from 50 to 100%, with an average of 76%. For further investigation, we selected only those markers which showed polymorphism. The 20 SSR markers in our study possessed 47 alleles, with an average of 2.35 alleles per locus. The PIC value ranged from 0.125 to 0.795, with an average of 0.424, which indicates that the SSR markers used in this study could be useful for genetic diversity studies onthe proso millet gene pool. Similar results were obtained for theproso millet germplasm using 25 polymorphic microsatellite markers [53]. Also, these results are in accordance with Minxuan Liu et al.; according to their research, a total of 179 alleles were detected, with an average of 2.7 alleles per locus, and the mean PIC and He were 0.376 and 0.445, respectively [54].

During the present study, to detect the correlations between nine polymorphic markers and the studied agronomic traits, at-test analysis was conducted. This study concluded that the SSR 85 and SSR 86 markers were associated with the PT and GY traits. These results could be used in marker-assisted selection for yield productivity, and support the idea that SSR markers can provide the fast detection of genes of interest.

Genetic diversity analysis based on SSR markers can provide insights into the origin and evolution of proso millet. We evaluated the population structure and differentiation of the 100 accessions of different origins using the STRUCTURE 2.2 software. The delta K analysis, including 100 accessions from six origin groups, suggested that the most likely number of clusters was at K = 5. Hierarchical levels in thepopulation structure could hardly be recognized at K = 2 and K = 3. Both clusters hadmixed groups, and no well-assigned population designations could therefore be recognized. In our study, a population structure analysis showed that accessions from the European, Central Asian and North Asian groups were mainly classified as having mixed populations. A dendrogram based on a neighbor-joining phylogenetic tree of the SSR data was constructed based onthe 100 proso millet accessions belonging to six origin groups. According to the UPGMA cluster analysis, all accessions were separated into three main clusters, but the genotypes from one origin did not form separate clusters. The genotypes of Central and North Asian origin that were investigated in this study showed broader genetic diversity. The PCoA based on the SSR genotyping of the 100 proso millet accessions was conducted using nine SSR markers. The accessions of the studied collection were divided into groups depending on their attribution to species and place of origin, respectively.

## 5. Conclusions

SSR marker-based evaluation is a very effective tool for assessing genetic diversity in proso millet genotype populations of different origins. In this research, 20 SSR markers were used to genotype 100 accessions of proso millet from different origins. A neighbor-joining phylogenetic tree divided the 100 proso millet accessions into three main clusters, where clusters 1 and 3 were mainly represented by Central Asian genotypes. The application of SSR markers suggested that Central and North Asian accessions have wide genetic differences from the other groups. Particularly, the principal coordinate plot showed that accessions from the Central and North Asian groups were distributed across the whole plot spectrum. The application of at-test indicated that SSR 85 and SSR 86 were associated with some agronomic traits, such as productive tillering (PT, pcs) and grain yield (GY, $g/m^2$). Thus, these marker trait associations can prove useful in proso millet breeding programs for the improvement of yield productivity.

**Author Contributions:** Conceptualization, A.O. and A.Z.; methodology, M.Z., I.Z. and G.Y.; investigation, G.Y. and A.R.; formal analysis, A.Z., I.Z. and A.R.; resources, E.D. and A.R.; writing—original draft preparation, E.D., A.R. and A.O.; writing—review and editing, M.Z., A.R. and E.D.; supervision, A.R. and E.D.; funding acquisition, A.R. All authors have read and agreed to the published version of the manuscript.

**Funding:** The work was carried out within the framework of the scientific project AP14870014,"Application of DNA technologies in breeding and genetic studies of millet culture when creating new domestic drought-resistant varieties" (2022–2024); grant funding for this research work was provided by the "Science Committee of the Ministry of Science and Higher Education of the Republic of Kazakhstan" State Institution.

**Institutional Review Board Statement:** Not applicable.

**Data Availability Statement:** Not applicable.

**Acknowledgments:** This work was supported by the RUDN University Strategic Academic Leadership Program.

**Conflicts of Interest:** The authors have no conflicts of interest to declare.

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
