# Peer review of "Microsatellite-Based Genetic Diversity Analysis and Population Structure of Proso Millet (Panicum miliaceum L.) in Kazakhstan"

_agronomy, doi:10.3390/agronomy13102514_

Round 1

Reviewer 1 Report

The paper “SSR Based Genetic Diversity Analysis and Population Structure of Proso millet (Panicum miliaceum L.) in Kazakhstan‘’  aimed to use SSR markers  to assess the polymorphism and genetic diversity of 100 millet accessions from different  countries, which were tested in the dry steppe zone conditions of the Akmola region from 2020 to 16 2022. 

The paper is prepared professionally. It includes a well-crafted abstract and an exhaustive introduction that justifies the research undertaken. The introduction points to the deficiencies in the literature on the subject. The aim is clearly defined. Modern analytical methods were used in the research. The discussion of the results is well prepared. The conclusions are well-defined. The illustrative material is appropriate.

In my opinion, the manuscript after corrections, will be suitable for publication in a journal.

Detailed comments:

Title: Do not use abbreviation (SSR) on title.

Abstract: Should include some more numeric data obtained from the study 

Do not use abbreviations when use first time.

Introduction - The introduction is enough in my opinion. Introduction needs some minor changes

Line 59-60 Assessment of genetic diversity in genbank collections has significance for plant breeding programs and genetic resources conservation [19, 20]. 

Please use more references to support this idea. I suggest below fresh references

Yildiz, M.; Arbizu, C. Inter-primer binding site (iPBS) retrotransposon markers provide insights into thegenetic diversity and population structure of carrots (Daucus, Apiaceae). Turk. J. Agric. For. 46 (2): 214 223. https://doi.org/10.55730/1300-011X.2972.

Daler, S.; Cangi, R. Characterization of grapevine (V. Vinifera L.) varieties grown in Yozgat province (Turkey) by simple sequence repeat (SSR) markers. Turk. J. Agric. For. 46 (1): 38-48.https://doi.org/10.3906/tar-2104-75.

Chen, J.D.; Zheng, C.; Ma, J.Q.; Jiang, C.K.; Ercisli, S.; Yao, M.Z.; Chen, L. The chromosome-scale genome reveals the evolution and diversification after the recent tetraploidization event in tea plant. Hortic Res. 2020 1;7:63. doi: 10.1038/s41438-020-0288-2. 

There is no Figure 1 in manuscript?????

Line 127-128 Give more information about SSR markers???

Genotyping proso millet germplasm was performed using a set of twenty (20) microsatellite or SSR markers (Table 2).

NA

Author Response

Dear Reviewer

We gratefully acknowledge the detailed revision of the text and useful suggestions to improve the paper by the reviewers. We have closely followed he/she suggestions and introduced the required changes in the text. Main changes are highlighted into the manuscript in YELLOW. Below, we have included reviewer comments and our responses.

Title: Do not use abbreviation (SSR) on title.

In the title of the paper the "SSR" word was changed to "Microsatellite".

Abstract: Should include some more numeric data obtained from the study

The abstract contains brief numeric data.

Do not use abbreviations when use first time.

Text revised.

Introduction - The introduction is enough in my opinion. Introduction needs some minor changes

Introduction revised.

Line 59-60 Assessment of genetic diversity in genbank collections has significance for plant breeding programs and genetic resources conservation [19, 20].

Please use more references to support this idea. I suggest below fresh references

Yildiz, M.; Arbizu, C. Inter-primer binding site (iPBS) retrotransposon markers provide insights into thegenetic diversity and population structure of carrots (Daucus, Apiaceae). Turk. J. Agric. For. 46 (2): 214 223. https://doi.org/10.55730/1300-011X.2972.

Daler, S.; Cangi, R. Characterization of grapevine (V. Vinifera L.) varieties grown in Yozgat province (Turkey) by simple sequence repeat (SSR) markers. Turk. J. Agric. For. 46 (1): 38-48.https://doi.org/10.3906/tar-2104-75.

Chen, J.D.; Zheng, C.; Ma, J.Q.; Jiang, C.K.; Ercisli, S.; Yao, M.Z.; Chen, L. The chromosome-scale genome reveals the evolution and diversification after the recent tetraploidization event in tea plant. Hortic Res. 2020 1;7:63. doi: 10.1038/s41438-020-0288-2.

We are agreed with Reviewer comments, and added all references which recommended for us.

There is no Figure 1 in manuscript?????

The figure 1 added.

Line 127-128 Give more information about SSR markers???

Genotyping proso millet germplasm was performed using a set of twenty (20) microsatellite or SSR markers (Table 2).

We corrected the title of Table 2 and added information about annealing temperature of SSR markers

We hope that after these enhancements the manuscript can now be accepted, although we are certainly willing to consider further changes if necessary.

Yours sincerely,

Reviewer 2 Report

This paper provides information on the genetic diversity of the important cereal plant Panicum miliaceum. Without dwelling on the positive aspects of the paper, it is necessary to draw the authors' attention to significant shortcomings.

1. The article contains a number of technical errors, starting with the first lines (lines 4-5). The entire text needs to be thoroughly revised to remove the technical flaws and to include the figures (Fig. 1 is not included in the manuscript).

2. What is the actual ploidy level of Panicum miliaceum? Tetraploid (line 33) or diploid (line 35)?

3. Information on the origin of all studied accessions must be provided (in an appendix or in supplementary materials). This information is very important for other researchers and for the overall interpretation of the results. 

4. Detailed methodology of the tests and experiments must be provided (lines 98-101). Simply linking to a website where this information is not available or is in Russian and not accessible to a large part of the readership [41] is not sufficient.

5. Which agronomic traits were assessed (lines 102-103)? To be specified in detail in the methodology. The number of samples used in the study, how the tests were carried out and other information on the evaluation of the plants must be included in the methodology. How was the weight of the seeds in the panicle assessed? How was the weight of 1000 seeds estimated, etc.? 

6. Why is there no mention of the various subspecies of Panicum miliaceum throughout the article? Have they not been analysed? For example, the globally widespread weedy subsp. ruderale could be of great importance for breeding. And has the colour of the grain husk, which is a very variable feature of these plants, been assessed? 

7. I missed the discussion and analysis in the article of the correlation between the traits evaluated and the results of the genetic diversity assessment. What is the point of including information in an article that is not interlinked? However, if you are analysing traits and diversity, you should link these results. 

8. The discussion and conclusions need revision. The discussion is, in my opinion, very superficial. It does not discuss the productivity of the accessions evaluated. The conclusions need to be more focused and do not repeat what has already been said.

9. There are many formatting and other technical errors in the references. They need to be reformatted according to journal requirements. 

The language of the manuscript is satisfactory, but there are many technical errors (no spaces between words).

Author Response

Dear Reviewer

We gratefully acknowledge the detailed revision of the text and useful suggestions to improve the paper by the reviewers. We have closely followed he/she suggestions and introduced the required changes in the text. Main changes are highlighted into the manuscript in YELLOW. Below, we have included reviewer comments and our responses.

  1. The article contains a number of technical errors, starting with the first lines (lines 4-5). The entire text needs to be thoroughly revised to remove the technical flaws and to include the figures (Fig. 1 is not included in the manuscript).

All technical errors in the text corrected

  1. What is the actual ploidy level of Panicum miliaceum? Tetraploid (line 33) or diploid (line 35)?

In the paper we corrected: “Millet is an allotetraploid cereal (2n =4×= 36)”

  1. Information on the origin of all studied accessions must be provided (in an appendix or in supplementary materials). This information is very important for other researchers and for the overall interpretation of the results.

Information on the origin of all studied accessions provided in supplementary materials

  1. Detailed methodology of the tests and experiments must be provided (lines 98-101). Simply linking to a website where this information is not available or is in Russian and not accessible to a large part of the readership [41] is not sufficient.

Added brief information about sowing information

  1. Which agronomic traits were assessed (lines 102-103)? To be specified in detail in the methodology. The number of samples used in the study, how the tests were carried out and other information on the evaluation of the plants must be included in the methodology. How was the weight of the seeds in the panicle assessed? How was the weight of 1000 seeds estimated, etc.?

Details in the methodology of 1000 weight seeds and number of seeds was added.

  1. Why is there no mention of the various subspecies of Panicum miliaceum throughout the article? Have they not been analysed? For example, the globally widespread weedy subsp. ruderale could be of great importance for breeding. And has the colour of the grain husk, which is a very variable feature of these plants, been assessed?

In the present study we worked only subspecies ssp. miliaceum, detailed information of taxonomic classification provided in supplementary materials. We have not analyzed the colour of the grain husk.

  1. I missed the discussion and analysis in the article of the correlation between the traits evaluated and the results of the genetic diversity assessment. What is the point of including information in an article that is not interlinked? However, if you are analysing traits and diversity, you should link these results.

Added information about correlation between SSR markers and main agronomic traits (Line 342-345).

  1. The discussion and conclusions need revision. The discussion is, in my opinion, very superficial. It does not discuss the productivity of the accessions evaluated. The conclusions need to be more focused and do not repeat what has already been said.

In the discussion made some revision

  1. There are many formatting and other technical errors in the references. They need to be reformatted according to journal requirements.

The language of the manuscript is satisfactory, but there are many technical errors (no spaces between words).

All technical errors in the references and in the paper are reformatted according to journal requirements

We hope that after these enhancements the manuscript can now be accepted, although we are certainly willing to consider further changes if necessary.

Yours sincerely,